# Impact of Neurons on Patient-Derived Cardiomyocytes Using Organ-On-A-Chip and iPSC Biotechnologies

**DOI:** 10.3390/cells11233764

**Published:** 2022-11-25

**Authors:** Albin A. Bernardin, Sarah Colombani, Antoine Rousselot, Virginie Andry, Yannick Goumon, Hélène Delanoë-Ayari, Côme Pasqualin, Bernard Brugg, Etienne D. Jacotot, Jean-Luc Pasquié, Alain Lacampagne, Albano C. Meli

**Affiliations:** 1PhyMedExp, University of Montpellier, Inserm, CNRS, 371 Avenue du Doyen G. Giraud, CEDEX 5, 34295 Montpellier, France; 2MicroBrain Biotech S.A.S., 78160 Marly Le-Roi, France; 3Université Paris-Saclay, CEA, CNRS, NIMBE, 91191 Gif-sur-Yvette, France; 4SMPMS-INCI, Mass Spectrometry Facilities of the CNRS UPR3212, CNRS UPR3212, Institut des Neu-Rosciences Cellulaires et Intégratives, Centre National de la Recherche Scientifique and University of Strasbourg, 68009 Strasbourg, France; 5Claude Bernard University, Université de Lyon, Institut lumière matière, 69000 Lyon, France; 6Groupe Physiologie des Cellules Cardiaques et Vasculaires, Université de Tours, EA4245 Transplantation, Immunologie, Inflammation, 37000 Tours, France; 7Sorbonne Université, Campus Pierre et Marie Curie, Institut de Biologie Paris-Seine, CNRS UMR 8256, INSERM U1164, F-75005 Paris, France; 8The Taub Institute for Research on Alzheimer’s Disease and the Aging Brain, Columbia University, New York, NY 10032, USA; 9Department of Pathology and Cell Biology, Vagelos College of Physicians and Surgeons, Columbia University, New York, NY 10032, USA; 10Department of Cardiology, Montpellier University Hospital, 34295 Montpellier, France

**Keywords:** organ-on-a-chip, microfluidic system, iPSC, cardiomyocytes, neurons

## Abstract

In the heart, cardiac function is regulated by the autonomic nervous system (ANS) that extends through the myocardium and establishes junctions at the sinus node and ventricular levels. Thus, an increase or decrease in neuronal activity acutely affects myocardial function and chronically affects its structure through remodeling processes. The neuro–cardiac junction (NCJ), which is the major structure of this system, is poorly understood and only a few cell models allow us to study it. Here, we present an innovant neuro–cardiac organ-on-chip model to study this structure to better understand the mechanisms involved in the establishment of NCJ. To create such a system, we used microfluidic devices composed of two separate cell culture compartments interconnected by asymmetric microchannels. Rat PC12 cells were differentiated to recapitulate the characteristics of sympathetic neurons, and cultivated with cardiomyocytes derived from human induced pluripotent stem cells (hiPSC). We confirmed the presence of a specialized structure between the two cell types that allows neuromodulation and observed that the neuronal stimulation impacts the excitation–contraction coupling properties including the intracellular calcium handling. Finally, we also co-cultivated human neurons (hiPSC-NRs) with human cardiomyocytes (hiPSC-CMs), both obtained from the same hiPSC line. Hence, we have developed a neuro–cardiac compartmentalized in vitro model system that allows us to recapitulate the structural and functional properties of the neuro–cardiac junction and that can also be used to better understand the interaction between the heart and brain in humans, as well as to evaluate the impact of drugs on a reconstructed human neuro–cardiac system.

## 1. Introduction

The cardiac function is tightly regulated by the autonomic nervous system (ANS) that extends through the myocardium [1,2]. The autonomic nervous system is composed of the parasympathetic nervous system (PNS) and sympathetic nervous system (SNS) through the establishment of the neuro–cardiac junction (NCJ) at the sinus node and ventricular levels [1,2,3,4]. The activity of the ANS and the regulation of cardiac activity is under the control of the medulla oblongata. During emotional or physical stress, electrical signals from the body and the cortical nervous system reach the neurons localized in the different regions of the medulla oblongata, which in turn activate neurons of the sympathetic or parasympathetic systems. NCJs allow the interaction of ANS neurons and cardiomyocytes for rapid regulation of homeostasis and heart integrity. The ANS also exerts a trophic action during the development [5,6] and under cardiac pathologies [4]. Thus, an increase or decrease in sympathetic activity directly affects myocardial function and its structure through remodeling processes.

Unlike the neuromuscular junctions, synapses of the neuro–cardiac junction are poorly understood. A few rare studies have demonstrated the presence of protein complexes involved in cellular interaction or in exocytosis [7]. Immunostaining has revealed a specialization of the cardiomyocyte membrane, with a large concentration of β1-adrenergic receptors, and a decreased expression of caveolin-3.

The secretion of noradrenalin is involved in the development of the myocardium by the tropism it exerts [8]. Dowell has demonstrated its involvement during myocardial development. The specific death of sympathetic neurons by the addition of 6-hydroxydopamine is directly correlated with a decrease in myocardial size [5]. Kreipke and Birren showed that noradrenalin is involved in the transition between hyperplastic and hypertrophic stages [6] and directly impacts the size and number of cardiomyocytes (CMs).

Several heart conditions are associated with changes in the intracardiac neuronal network that can lead to cardiac arrhythmias or an increased risk of myocardial infarction. These attacks can be induced by neuronal degeneration associated with diabetes [9], sympathetic hyper-innervation [10] or infection of the stellate ganglia [11] as the major sympathetic ganglion for cardiac regulation [12].

There are very few models with which to study the NCJ. The cellular models are based on the use of direct co-cultures in the same compartment [7,13,14]. Those models involve mixing the cells together in non-optimal conditions with the lack of control on specific cell population. The development of microfluidic systems allows the establishment of hybrid co-culture and organ-on-a-chip (OOC) with the possibility of cultivating different cell types in separate compartments connected by micro-channels [15]. Spatial and fluidic compartmentalization of distal axons and cell bodies replicates the in vivo conditions. Somatic and axonal environments are independently controlled. Compartmentalized devices are used in a variety of experimental models ranging from isolating axons to forming neuronal networks, from creating axon co-cultures to engineering axonal behavior. Such protocols have emerged in several areas of research such as angiogenic development [16] or extravasation processes in tumors [17]. Other groups have developed similar systems to study the NCJ [13,18,19], demonstrating its feasibility. The human-induced pluripotent stem cells (hiPSC) are self-renewing and have the potential to differentiate into any somatic cells. They nowadays offer great opportunities to create patient-derived OOC and model key physiological and pathophysiological features of human organs [20]. Such cellular models are promising for studying the physiology and structure of the NCJ during cardiac development and regulation to better understand the mechanisms involved during the establishment of pathologies. This platform can also be used for testing new compounds to restore cardiac or neuronal properties under pathological conditions.

In the present study, we designed an innovant neuro–cardiac OOC based on microfluidic chip and hiPSC to study the impact of rat sympathetic neurons (differentiated for PC12 cells) or human hiPSC-derived autonomic neurons (hiPSC-NRs) on the functional properties of ventricular-like human hiPSC-derived cardiomyocytes (hiPSC-CMs).

## 2. Materials and Methods

### 2.1. Microfluidic Devices

Microfluidic chips were produced by standard molding methods using epoxy-based negative photoresists (SU-8) and MicroBrain Biotech proprietary microdesigns (Brainies™, Cat#: MBBT5; Marly le Roi, France). Briefly, Polydimethylsiloxane (Sylgard 184, PDMS; Dow Corning) was mixed with curing agent (9:1 ratio) and degassed under vacuum. The resulting preparation was poured onto a chosen SU8 mold and reticulated at 70 °C for 2 at least hours. The elastomeric polymer print was detached, and 2 reservoirs were punched for each chamber. The polymer print and a glass cover slip were cleaned with isopropanol, dried, and treated for 3 minutes in an air plasma generator (98% power, 0.6 mBar, Diener Electronic) and bonded together. Brainies^®^ MBBT5 is a chip with a design containing 4 neuronal diodes. One neuronal diode includes 2 rectangular culture chambers (volume ~1 µL) each connected to 2 reservoirs and separated by a series of 500 µm-long asymmetrical micro-channels (3 µm high, tapering from 15 µm to 3 µm).

### 2.2. Rat PC12 Culture 

Rat adrenal gland (phaeochromocytoma) PC12 cells from the European Collection of Authenticated Cell Cultures (ECACC, Salisbury, UK, 88022401) were seeded on a collagen-coated 60 mm Petri dish in complete medium. Complete medium was composed of RPMI 1640 (Gibco, Waltham, MA, USA, 21875091) supplemented with 10% horse serum (Gibco, Waltham, MA, USA, 26050088), 5% Fetal Bovine Serum (Gibco, Waltham, MA, USA, 26140079) and 12.5 µg/mL gentamicin (Sigma, St. Louis, MO, USA, G1397). PC12 differentiation was induced by switching from complete medium to differentiation medium. Differentiation medium was composed of RPMI 1640 supplemented with 1% horse serum, 12.5 µg/mL gentamicin and 50 ng/mL nerve growth factor (NGF, Aachen, Germany, ABIN804475).

### 2.3. hiPSCs Maintenance and Differentiation

hiPSC cell lines were obtained from healthy male control. hiPSC were maintained in StemFlex medium (Gibco, Waltham, MA, USA, A3349401) on Matrigel hESC-qualified matrix (Corning, NY, USA, 354277)-coated Petri dishes at 37 °C in 5% CO2 incubator between passage 17 to 27. When the cells reached confluence, cell monolayer was dissociated using Tryple express enzyme (Gibco, Waltham, MA, USA, 12604013) to obtain single cell solution. We differentiated patient-specific hiPSC-derived cardiomyocytes as previously published [21,22]. hiPSC were differentiated in ventricular-like cardiomyocytes using monolayer protocol and sandwich approaches [23]. The hiPSC-CMs were purified using the lactate-based protocol in glucose- and pyruvate-free conditions supplemented with 4 mM Na-lactate as previously [24].

The hiPSC-derived autonomic neurons were obtained from the same cell-line and differentiated following protocols described by Montgomery group [25]. Neuronal induction was launched on a cell monolayer cultured on culture dishes coated on Matrigel hESC-qualified matrix. Neuronal induction was conducted for 12 days to obtain a population of neurons. These neurons are seeded in microfluidic chips and matured for 42 days in co-culture with hiPSC-CMs.

### 2.4. Neurotransmitter Release Quantification

The acetylcholine neurotransmitter, released in cell medium was quantified using fluorometric kit (Acetylcholine, Abcam, Waltham, MA, USA, ab65345). The four neurotransmitters were quantified in supernatant of 14 days differentiated PC12 cells harvested in conditioned medium after addition of Carbachol (1.5 mM) in the medium.

Mass spectrometry was then performed. The cell secretion media were collected. Cells were recovered in 50 µL of H_2_O containing 0.1 mM ascorbic acid. Cells were sonicated (4 × 10 s, 100 W; Fisher Scientific, Waltham, MA, USA; Model 505 Sonic Dismembrator). After centrifugation (20,000× *g*, 30 min, 4 °C), supernatants were collected and protein concentrations were accessed using the Protein Assay kit (Bio-Rad). An isotopic dilution approach was used for the absolute quantification. Either 20 µL of the cell extracts or 50 µL of the secretion medium were mixed with 10 µL of internal standards containing 20 pM of D4-Dopamine, C6-Noradrenalin, D6-Adrenalin, L-DOPA in 0.1 mM ascorbic acid1. Then, 40 µL of borate buffer and 10 µL AccQtag Ultra reagent (AccQ-Tag Ultra derivatization kit, Waters, Guyancourt, France) were added to cell extracts and secretion media. The mixture was incubated 10 min at 55 °C under agitation. Then, 500 µL of ice-cold acetonitrile (ACN) were added and samples were centrifuged (20,000× *g*, 30 min, 4 °C). The resulting supernatants were dried under vacuum and suspended in 20 µL of H_2_O containing 0.1% formic acid (*v*/*v*).

Analyses were performed on a Dionex Ultimate 3000 HPLC system (Thermo Scientific, Waltham, MA, USA) coupled with an Endura triple quadrupole mass spectrometer (Thermo Electron, Waltham, MA, USA). The system was controlled by Xcalibur v. 2.0 software (Thermo Electron). Next, 5 µL of samples were loaded into reverse phase Zorbax column (#863600-902; 1 mm × 150 mm, 3.5 µm, Agilent Technologies, Santa Clara, CA, USA, SB-C18). Elution of the compounds was performed at a flow rate of 90 µL/min, at 40 °C (see Appendix A for conditions). Buffer A corresponded to H_2_O 98.9%/ACN 1%/formic acid 0.1% (*v*/*v*/*v*) and buffer B was ACN 99.9%/formic acid 0.1% (*v*/*v*). Dopamine, adrenalin and noradrenalin are measured using the multiple reaction monitoring mode (MRM) according to the settings. The targeted compounds are detailed in Appendix A. The selection of the monitored transitions and the optimization of the collision energy (CE) were manually determined. The identification of the compounds was based on precursor ions, daughter ions and retention times obtained for dopamine, adrenalin and noradrenalin and their corresponding internal standards. Amounts of neurotransmitters were quantified according to the isotopic dilution method [26].

### 2.5. PC12/hiPSC-CMs Co-Culture

The day before neuronal seeding, chips were UV-sterilized for 20 min, then coated with a solution of poly-D-lysine (10 µg/mL, Sigma, St. Louis, MO, USA, P6407), incubated overnight (37 °C, 5% CO_2_), and rinsed 3 times with Dulbecco’s phosphate-buffered saline (PBS) (Sigma, St. Louis, MO, USA, D8537). Then, 4 h before cell seeding, chips were treated with a solution of Laminin (10 µg/mL: Sigma, St. Louis, MO, USA, L2020) in PBS. Undifferentiated PC12 cells were seeded in microfluidic devices in differentiation medium supplemented with 50 ng/mL of NGF and differentiated for 5 days. During this phase, PC12 started to differentiate with the formation of neurites and started projecting into micro-channels and invaded cardiac compartment. At day 5, hiPSC-CMs aged between 11 days and 20 days were seeded on Matrigel hESC-qualified matrix with a density of 20,000 cells/compartment in B27 medium supplemented with 10 µM Rock inhibitor and NGF at 150 ng/mL. Complete medium changes with corresponding NGF concentrations were performed once a week to avoid cell detachment.

For neuro–cardiac OOC between hiPSC-NRs and hiPSC-CMs, autonomic neurons are seeded in neuronal compartments on Matrigel hESC-qualified and matured for 7 days before cardiomyocytes seeding without addition of NGF. Once the cardiomyocytes seeded, co-culture started for 30 days to allow neurites to invade cardiac compartment and create interaction with CMs.

### 2.6. Immunocytochemistry

Immunocytochemistry (ICC) were performed 14 days after cells seeding on chips. Classical ICC protocols have been adapted to microfluidic chips to avoid liquid flux and cell detachment. Cells in neuronal and cardiac compartments were fixed with 4% PFA for 15 min at room temperature then rinsed 3 times with PBS. Cell membranes were permeabilized with 0.1% Triton for neuronal cells and 0.01% Triton for cardiac cells in PBS, 10 min at room temperature followed by blockade solution (1% BSA in PBS) for 30 min at room temperature. Primary antibodies were diluted in PBS supplemented with 0.1% BSA and incubated at 4 °C overnight and rinsed three times with PBS. Secondary antibodies were diluted in PBS and 0.1% BSA and incubated for 2 h at room temperature supplemented with 40,6-diamidino-2-phenylindole (DAPI, 1:2000). Antibodies are listed in Appendix A. The chips were stored hydrated at 4 °C before use. Staining was obtained with inverted microscope Axio Observer Z1 coupled with laser scanning confocal microscope LSM800 piloted with Zen 2.5 software, all provided by Zeiss. ICC images were acquired in 3D with z-stack option and reconstructed with ImageJ.

### 2.7. Intracellular Calcium Dynamics

Intracellular calcium signaling was studied to confirm the modulation of cardiac functional properties after neuronal stimulation. Calcium transient (CaT) were recorded with cytosolic free calcium probe Fluo-4 AM (Molecular Probes™, Eugene, OR, USA, F14201) as published before [27,28]. Cells were incubated with 1 µM probe for 15 minutes at 37 °C in Tyrode solution (in mM: 135 NaCl, 4 KCl, 1.8 CaCl_2_, 1 MgCl_2_, 10 HEPES, 10 glucose, 0.1 acid ascorbic, pH 7.4). Calcium transients were recorded for 7 s in line-scan mode using LSM800 confocal microscope (Zeiss) and analyzed using Peakinspector as before [21,22,29].

### 2.8. Measurement of Contractile Function Using Video-Edge Capture

To assess the contractile properties of the hiPSC-CMs connected with the neurons, we performed the video-edge capture as before [21,22] on microfluidic chip after 14 days of co-culture. Activity of connected cardiomyocytes were recorded for 25 s. Transmitted light video at 63 frame/s (fps) were obtained with LSM800 confocal microscope (Zeiss) at ×63 magnification with ORCA-Flash4 camera (Hamamatsu). Experiments were performed in Tyrode solution (in mM: 135 NaCl, 4 KCl, 1.8 CaCl_2_, 1 MgCl_2_, 10 HEPES, 10 glucose, 0.1 acid ascorbic, pH 7.4). The movement of each pixel in the image was evaluated using a patented video analysis script (Autobeats) via MATLAB software [21,22]. This analysis provides a contraction versus time curve from which contractile properties are extracted.

### 2.9. Statistical Analysis

All data are expressed as the mean ± SEM. Statistical analysis was performed with GraphPad Prism 8.0.1 (GraphPad Software Inc., San Diego, CA, USA, Prism8). Experimental groups were compared by using non-parametric Kruskal–Wallis tested followed by Dunn’s multiple comparisons test were performed. *p* < 0.05 was considered statistically significant.

## 3. Results

### 3.1. Molecular Characterization of the PC12 Neurons Connected with hiPSC-CMs

Rat PC12 cells were differentiated for 14 days in a classic Petri dish. Neurotransmitter production was confirmed by ICC showing the expression of enzymes involved in neurotransmitter production: choline acetyltransferase (ChAT) for acetylcholine, tyrosine hydroxylase for all the catecholamines (TH) and dopamine β-hydroxylase (DBH) for noradrenalin. ICC showed that PC12 expressed both TH and ChAT in neurites and DBH in their soma (Figure 1A), suggesting the production of these neurotransmitters. Despite these staining, we were unable to measure catecholamines with ELISA kits (Abnova, KA3836), in contrast to acetylcholine, which could be measured with fluorometric kits (Abcam, ab65345). The neurotransmitter quantification was performed in cell supernatant after addition of 1.5 mM Carbachol. Medium was collected 2 min after neuronal stimulation. Our results showed a larger secretion of acetylcholine (24.39 10^3^ pM) than catecholamines or their precursors (noradrenalin: 3.7 pM, L-DOPA: 3.8 pM, dopamine: 214.6 pM) in PC12 (Figure 1B), suggesting a dominant parasympathetic profile of these neurons.

As published before [21,22,30], we confirmed that we generated ventricular-like hiPSC-CMs through the expression of the sarcomeric cardiac markers including α-actinin and cardiac troponin I (cTnI) at 30 days of differentiation (Figure 1C).

### 3.2. Microfluidic Devices Allow a Neuro–Cardiac OOC

Twenty years ago, Taylor et al. used a microfabricated silicon wafer to mold a silicone elastomer (namely polydimethylsiloxane) that featured two chambers connected by parallel microgrooves. Once sealed against a glass coverslip, the microgrooves formed microchannels [31,32]. The resulting devices were developed in many ways by multiple research groups to compartmentalize and study neurons as well as neuronal networks in various physiological and pathological contexts [33].

Neuronal diodes are fluidic microsystems, inspired by those of Taylor et al., conceived to impose directionality when connecting two neuronal populations [34]. Neuronal diodes are made up of two separate cellular micro-compartments interconnected by a series of asymmetric microchannels, each of whose openings are of different sizes (Figure 2A). Here, we have used a slightly modified and standardized design of the two-chamber neuronal diode (Figure 1; Brainies^®^ MBBT5) and seeded neurons in the compartment where the microchannels openings were widest (15 µm) and cardiomyocytes in the second compartment, where the openings were very narrow (3 µm). The expectation of such cocultures is to separate the (somato-dendritic) neuronal compartment from the cardiomyocytes compartment, and to obtain axonal projections through the microchannel to analyze axon-dependent neuron to cardiomyocytes interactions. The characteristics of the device combined with the controlled filling of each reservoir allow fluidic isolation between the chamber and the so-called compartmentalized pharmacology where treatment can be added selectively in one chamber but not the opposite.

### 3.3. PC12/hiPSC-CMs OOC Involves Neuro–Cardiac Junctions and Synapse-Like Junctions

Undifferentiated PC12 cells were seeded in neuronal compartments with 50 ng/mL of NGF. The cells started differentiating in neuron-like cells through neurites from the soma. NGF gradient concentration was created between neuronal (50 ng/mL) and cardiac compartment (150 ng/mL) to attract neurites. At day 5 after PC12 seeding, we observed neurites projecting into the microchannel and starting to invade the cardiac compartment (Figure 2B). hiPSC-CMs were seeded on this day and OOC were co-cultured for 14 days. This procedure allowed us to obtain reproductible and transposable co-cultures protocols. During the 14 days of co-culture, PC12 neurites continued to develop in cardiac compartments and projected in every direction to create numerous interactions with cardiomyocytes (Figure 2C). At the end of the microchannels, PC12 projections were organized in a network of large neurites. This network divided into several smaller neurites systems that projected in all directions and interacted with the hiPSC-CMs.

Previous studies have shown that the neurocardiac junction is a specialized structure that exhibits a specific cellular organization of neuronal and cardiac cells [7,14]. In vivo, sympathetic neurons project along cardiac tissue and form buds with cardiomyocytes. This morphology is referred to as the pearl necklace structure [35]. These axonal sprouting are also called varicosity, which contain the cardiac synapse releasing the neurotransmitter. Due to their secretory activity and interaction with cardiomyocytes, varicosities express a panel of proteins involved in the release of neurotransmitters such as synapsin-1 [7] or proteins involved in intercellular interaction such as SNARE proteins [36].

To confirm the presence of synapses between PC12 neurites and hiPSC-CMs, we performed immunofluorescence experiments and acquired images in transmission light. Our results revealed that PC12 neurites interacted densely with hiPSC-CMs and formed varicosity-like structures (Figure 2D and Figure 3A). To confirm the presence of synapses, we revealed the expression of synapsin-1 and β3-tubulin following 14 days of co-culture using fluorescence microscopy. Our immuno-staining showed that neurites reproduced pearl necklace organization close to the hiPSC-CMs with the specific expression of synaspin-1 localized in the buds (Figure 3B). Notably, we detected no synapsin-1 and no TH in hiPSC-CMs seeded alone in the OCC (see Appendix A).

### 3.4. Neurons Modulate the Intracellular Calcium Handling in hiPSC-CMs

We then hypothesized that PC12 neuronal cells could modulate the ECC of the hiPSC-CMs via NCJ. We focused on the intracellular Ca^2+^ handling and contractile properties as two major aspects of the ECC. We designed a protocol to apply drugs independently in the neuronal and cardiac compartments (Figure 4A). Using Fluo4-AM and fluorescent microscopy, we acquired CaT at rest and in presence of Carbachol (muscarinic receptor agonist) in the neuronal compartment or 1 µM Atropine (muscarinic receptor antagonist) in the cardiac compartment (Figure 4B). Their effects were studied just after the addition of drugs in the corresponding compartments. We focused on the relative CaT amplitude as indicator of the sarcoplasmic reticulum (SR) Ca^2+^ content, velocity of Ca^2+^ release through type 2 ryanodine receptor (RyR2), Ca^2+^ reuptake duration and CaT frequency (Figure 4C).

Release of the neurotransmitter after the stimulation of PC12 cells by Carbachol in neuronal compartment affects the Ca^2+^ kinetics in cardiomyocytes (Figure 4D–F). The CaT frequency (Rest: 0.48 ± 0.04 Hz, Carbachol: 0.38 ± 0.03 Hz, Atropine: 0.63 ± 0.09 Hz, Atropine + Carbachol: 0.53 ± 0.07 Hz, *p* = ns) was not affected by the different experimental conditions in cardiomyocytes (Figure 4F). The addition of Carbachol did not modulate the CaT amplitude (Rest: 2.28 ± 0.13, Carbachol: 2.03 ± 0.14, *p* = ns, Figure 4C) but affected Ca^2+^ release velocity after PC12 stimulation (Rest: 11.41 ± 0.86, Carbachol: 7.76 ± 0.73, *p* < 0.01, Figure 4D) and Ca^2+^ reuptake duration (Rest: 0.87 ± 0.04 s, Carbachol: 1.04 ± 0.07 s, *p* < 0.05, Figure 4E).

The blockade of the M2 muscarinic receptor through the application of Atropine in the cardiac compartment induced an increase in Ca^2+^ release velocity in hiPSC-CMs compared to rest condition (Rest: 11.41 ± 0.86, Atropine: 21.54 ± 3.72, *p* < 0.05), after the addition of Carbachol (Carbachol: 7.76 ± 0.73, Atropine: 21.54 ± 3.72, *p* < 0.001) and after the addition of both molecules combined (Atropine: 21.54 ± 3.71, Atropine + Carbachol: 14.14 ± 2.10, *p* < 0.05) (Figure 4D). The addition of Atropine also induced a decrease in Ca^2+^reuptake duration compared to rest (Rest: 0.87 ± 0.04 s, Atropine: 0.66 ± 0.05 s, *p* < 0.05) or to Carbachol-treated PC12 cells (Carbachol: 1.04 ± 0.07 s, Atropine: 0.660 ± 0.047 s, *p* < 0.001) or both combined (Carbachol: 1.042 ± 0.068 s, Atropine + Carbachol: 0.71± 0.05 s, *p* < 0.001) (Figure 4E). Finally, we also observed that the addition of Atropine significantly increased Ca^2+^ transient amplitude compared to rest (Rest: 2.28 ± 0.13, Atropine: 3.30 ± 0.28, *p* < 0.001) and to Carbachol condition (Carbachol: 2.09 ± 0.17, Atropine: 3.30 ± 0.28, *p* < 0.001) (Figure 4C). 

To exclude co-culture artefacts, we compared the kinetic properties of the intracellular calcium cycling in hiPSC-CMs cultivated alone or with PC12 for 14 days in the chips. We observed that PC12 cause an increase in the calcium release velocity, suggesting that PC12 increase the rising phase of the calcium transients (see Appendix A). Furthermore, in absence of PC12, we monitored the effect of Carbachol applied in the neuronal compartment while Atropine was applied in the cardiac compartment containing the hiPSC-CMs. We found that Atropine has no effect on the SR calcium handling kinetic properties. However, addition of Carbachol surprisingly increased the calcium transient amplitude. Carbachol did not change the calcium release velocity, calcium reuptake duration or calcium transient frequency (see Appendix A).

### 3.5. Neurons Do Not Impact the Contractile Properties in hiPSC-CMs

We next evaluated whether neurons could modulate the contractile properties of the hiPSC-CMs using the video-edge capture, as published before [21,22], in microfluidic chips in transmitted light at rest (Figure 5A) and in the presence of carbachol and/or atropine. The contractility was first measured at rest for 25 s, then after addition of carbachol or Atropine. An example of the obtained contraction/relaxation cycle is shown in Figure 5B.

Acetylcholine release from PC12 neurites following the addition of Carbachol has no effects on contraction parameters (Figure 5 C–H). The contraction frequency in beats/min (Rest: 16.28 ± 1.39 bpm, Carbachol: 17.85 ± 1.63 bpm, *p* = ns, Figure 5C), contractile amplitude (Rest: 0.57 ± 0.07, Carbachol: 0.54 ± 0.05, *p* = ns, Figure 5D), contraction velocity (Rest: 2.94 ± 0.34 m/s, Carbachol: 2.94 ± 0.39 m/s, *p* = ns, Figure 5E), relaxation velocity (Rest: 1.87 ± 0.24 m/s, Carbachol: 2.23 ± 0.44 m/s, *p* = ns, Figure 5F), contraction homogeneity (Rest: 1.68 ± 0.05, Carbachol: 1.79 ± 0.07, *p* = ns, Figure 5G) and the resting time (Rest: 5310 ± 663.6 ms, Carbachol: 5304 ± 1318 ms, *p* = ns, Figure 5H). Surprisingly, addition of Atropine had no effects on the contractile properties compared to the results obtained with calcium experiments (Figure 4C–F) suggesting that spontaneous acetylcholine release is not sufficient to modulate the contraction frequency (Rest: 16.28 ± 1.39 bpm, Atropine: 21.15 ± 2.34 bpm, Atropine + Carbachol: 20.05 ± 2.57 bpm, *p* = ns, Figure 5C), the amplitude of contraction (Rest: 0.57 ± 0.07, Atropine: 0.44 ± 0.02, Atropine + Carbachol: 0.48 ± 0.03, *p* = ns, Figure 5D), the contraction velocity (Rest: 2.94 ± 0.34 m/s, Atropine: 2.28 ± 0.22 m/s, Atropine + Carbachol: 2.85 ± 0.37 m/s, *p* = ns, Figure 5E), the relaxation velocity (Rest: 1.87 ± 0.24 m/s, Atropine: 1.79 ± 0.18 m/s, Atropine + Carbachol: 2.08 ± 0.25 m/s, *p* = ns, Figure 5F), the contraction homogeneity (Rest: 1.68 ± 0.05, Atropine: 1.73 ± 0.05, Atropine + Carbachol: 1.82± 0.05, *p* = ns, Figure 5G) and resting time (Rest: 5310 ± 663.6 ms, Atropine: 2880 ± 518.9 ms, Atropine + Carbachol: 3433± 1082 ms, *p* = ns, Figure 5H).

### 3.6. Patient-Specific Neuro–Cardiac OOC Using hiPSC-NRs and hiPSC-CMs

In order to confirm the feasibility of such neuro–cardiac OOC using the same hiPSC line to generate neurons and cardiomyocytes, we differentiated TH-positive hiPSC-NRs that we co-cultivated with hiPSC-CMs in microfluidic devices (Figure 6A). The hiPSC-NRs were seeded in microfluidic compartments at the beginning of neuronal maturation 1 week before cardiomyocytes seeding. They were able to create numerous interactions with the projection of other neurons (Figure 6C) and to project into microchannels to invade cardiac compartment without NGF addition (Figure 6B). These neurons generated numerous interactions with one or several cardiomyocytes after 14 days of co-culture (Figure 6D).

## 4. Discussion

In this study, we built-up a neuro–cardiac OOC to investigate the functional impact of neurons on patient-derived cardiomyocytes. OOC approaches emerge with different protocols in order to study the effect of interaction between cardiac and neurons with the use of direct co-culture [13,14,37,38] or microfluidic devices. Compared to direct co-culture, microfluidic chips allow the culture of each cell-lines in their optimal conditions. Fluidic isolation of each chamber also allows us to ensure cell-type-specific pharmacological treatments. Here, our results indicate that neurons seeded in one compartment are able to further differentiate and spread axons into microchannels to reach the cardiac compartment containing hiPSC-CMs. The width and length of microchannels only allows the axon to reach the second compartment, whereas neither dendrites and soma, nor cardiomyocytes can access the opposite compartment.

Our data show that axons form a large network at the end of microchannels. Via a large spread, they subdivide in thin filaments which spread in all the directions to create interaction with hiPSC-CMs. This axonal network is the central element of neurocardiac regulation [39]. In physiological conditions, it has been shown that neurites spread over long distances from one cardiomyocyte to another and interact with them by forming buddings along axons. These “*en passant*” structures, also called varicosities, are the functional neurotransmitter release site that enables rapid and axial neurotransmitter release to several cardiomyocytes at the same time [39]. As published by other groups, these varicosities are a specialized and ordered structure on either side of the synaptic cleft that induces rapid activation of the β-adrenergic signaling pathway [14]. In the neuromuscular junction, these structures are also observed with high concentration of nicotinic receptors at the post-synaptic side [40].

To confirm the development of a functional synapse between PC12-derived neurons and hiPSC-CMs, we performed structural and functional characterization of the neuro–cardiac junction. The NCJ and its proteins are not very well known. In our study, we focused on the presence of varicosities in the neuronal network. We confirmed that expression of synapsin-1, a key synapse protein, at the neuro–cardiac interaction zone. Synapsin-1 is a protein expressed in the neuronal synapse to organize vesicles pools, maturation and neurotransmitter release [36,41,42]. In our neuro–cardiac OOC, we found that some structures similar to varicosities co-localized with synapsin-1. These observations strongly suggest that PC12 axons and cardiomyocytes form an NCJ-like structure and synapses in vitro. Further experiments using electronic microscopy could confirm such a specialized structure by showing an organized cell membrane in neurons and cardiomyocytes.

Our findings revealed that PC12 cells express both adrenergic (DBH and TH) and cholinergic (ChAT) enzymes but secrete a larger amount of acetylcholine compared to catecholamine, resulting in a parasympathetic rather than sympathetic neurotransmitter pattern. This observed phenotype makes sense with our results on the intracellular Ca^2+^ handling properties we acquired. Using Carbachol and Atropine to activate and inhibit the muscarinic receptors, respectively, we monitored the impact on the CaT properties and found only slight consequences. We found that stimulation of PC12 cells by Carbachol modulates calcium release velocity and Ca^2+^ reuptake duration but did not affect the CaT of the hiPSC-CMs. Moreover, inhibition of M2 muscarinic receptors in hiPSC-CMs using Atropine causes increased CaT amplitude, Ca^2+^ release velocity via RyR2 and decreased Ca^2+^ reuptake duration via SERCA2a. These results indicated that acetylcholine is spontaneously secreted by PC12 neurites under non-stimulating conditions and binds to muscarinic receptors to modulate Ca^2+^ kinetics affecting the SR Ca^2+^ cycling in hiPSC-CMs. In the absence of PC12 neurites, Atropine had no effect on hiPSC-CMs while Carbachol alone in the neuronal compartment only affected the CaT amplitude, thus confirming the impact of differentiated PC12 neurons on the SR calcium handling in hiPSC-CMs. Our present results indicate that PC12 have an impact on the SR calcium handling not only limited to muscarinic receptor stimulation by acetylcholine. Very likely, the PC12 neurites release a mixture of neurotransmitters including acetylcholine, dopamine and other catecholamines that lead to pleiotropic consequences in the SR calcium handling properties.

Surprisingly, the addition of Carbachol or Atropine has no effect on the contractile properties compared to the results obtained with Ca^2+^ experiments, suggesting that spontaneous acetylcholine release is not sufficient to modulate contractility. This absence of effect is possibly due to excessive spontaneous secretion which would reduce the stock of the neurotransmitter. During stimulation of PC12 cells by Carbachol, this stock of neurotransmitter may be too low and insufficient to significantly impact the hiPSC-CMs contraction. This spontaneous secretion could also constitutively activate the cardiac receptors which could no longer respond to a new release of neurotransmitters. Another explanation could be that our neurocardiac OOC does not have enough interactions between the two cell types to allow a deep modulation of the contractile properties. Our data also suggest that PC12 cell may reveal limitations in studying the effect of the autonomic nervous system on cardiac function due to their abnormal neurotransmitter secretion phenotype and the co-expression of cholinergic and catecholaminergic enzyme.

The use of neurons derived from hiPSC is the next step. Similarly to other groups [13,37], here, we demonstrated the feasibility of neuro–cardiac OOC from hiPSC in microfluidic chips. Direct co-culture improves the maturity of peripheral neurons derived from hiPSC by improving their Ca^2+^ kinetics and the expression of peripheral marker such as TH and DBH [13]. The cardiac maturity was also improved by the increased expression of contractile protein α-actinin and proteins involved in Ca^2+^ signaling such as RyR2 and phospholamban [37]. Another group showed that a culture of CMs with cortical neurons both from hiPSC in microfluidic devices are able to create NCJ with limited functional effects [38]. These results show that without a pure autonomic neuronal population, we are limited in studying NCJ and their implications. However, these previous studies did not investigate the functional modulations induced by the neuronal innervation. With our OCC-based model, we are the first to show the modulation of cardiac calcium kinetics induced by neuronal stimulation, suggesting the presence of functional synapses. We also showed the presence of varicosities-like structures in areas where neuronal projections interact with cardiomyocytes.

In this context, two groups recently published ANS differentiation protocols to obtain sympathetic and/or cholinergic neurons from hiPSC. In these articles, they demonstrated that autonomic neurons are able to modulate the beating rate of cardiomyocytes co-cultured with them. Despite this important progress, they did not investigate whether autonomic neurons may modulate other functional properties such as calcium signaling and contractile properties. Moreover, it seems unclear if these neurons secrete noradrenalin or acetylcholine as desired. Further experiments will be needed to obtain specific hiPSC-NRs able to impact the hiPSC-CMs properties in healthy or disease-related conditions. Many physiological properties of the NCJ are still unknow and there is no dedicated model to study them. Neuro–cardiac OOC are powerful tools to study the interaction between ANS and the heart from their development to their involvement into cardiac pathologies. Neuro–cardiac OCC are also promising systems to evaluate the efficacy and safety of (new) pharmacological compounds able to act on autonomic neurons, cardiomyocytes or both.

## 5. Conclusions

In this work, we built-up a neuro–cardiac OOC allowing us to study the physiological interactions between the ANS and heart during development and pathologies. We revealed the presence of functional synapses between neurons and cardiomyocytes, in particular with synapsin-1, a protein involved in the release of neurotransmitter-containing vesicles and varicosities. Our neuro–cardiac OOC modulates the intracellular Ca^2+^ cycling of the patient-derived cardiomyocytes but does not change their contractile properties. Overall, this work provides evidence of the functionality of a human neurocardiac OOC to further study the relationship between the nervous system and heart in a dish.

## Figures and Tables

**Figure 1 cells-11-03764-f001:**
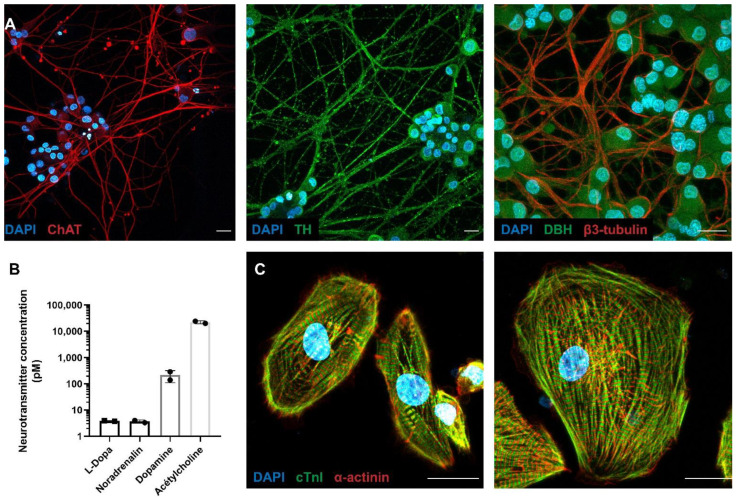
Molecular characterization of neuro–cardiac OOC composed of PC12 cells and hiPSC-CMs. (**A**) PC12 cells differentiate into parasympathetic neurons. PC12 were cultured for 14 days in the presence of 50 ng/mL NGF, fixed and immunolabelled for sympathetic markers. Representative immunofluorescence microscopy images show choline acetyltransferase (**left panel**; in red), tyrosine hydroxylase (**central panel**; in green), dopamine β-hydroxylase (**right panel**; in green), and β3-tubulin (**right panel**; in red). (**B**) Quantification of neurotransmitters in supernatant of PC12 cells. PC12 cells were differentiated for 14 days and subjected to addition of 1.5 mM Carbachol for 2 min. Then, medium was collected and the indicated neurotransmitters were quantified as described in the Materials and Methods section. (**C**) hiPSCs differentiate in cardiomyocytes (hiPSC-CMs) after 30 days following the so-called “matrix sandwich method” of differentiation. hiPSC-CMs were fixed and immunolabelled for sarcomeric cardiac markers. Representative immunofluorescence microscopy images show Cardiac troponin (in green) and the α-sarcomeric actinin (in red). Scale bars: 20 µm.

**Figure 2 cells-11-03764-f002:**
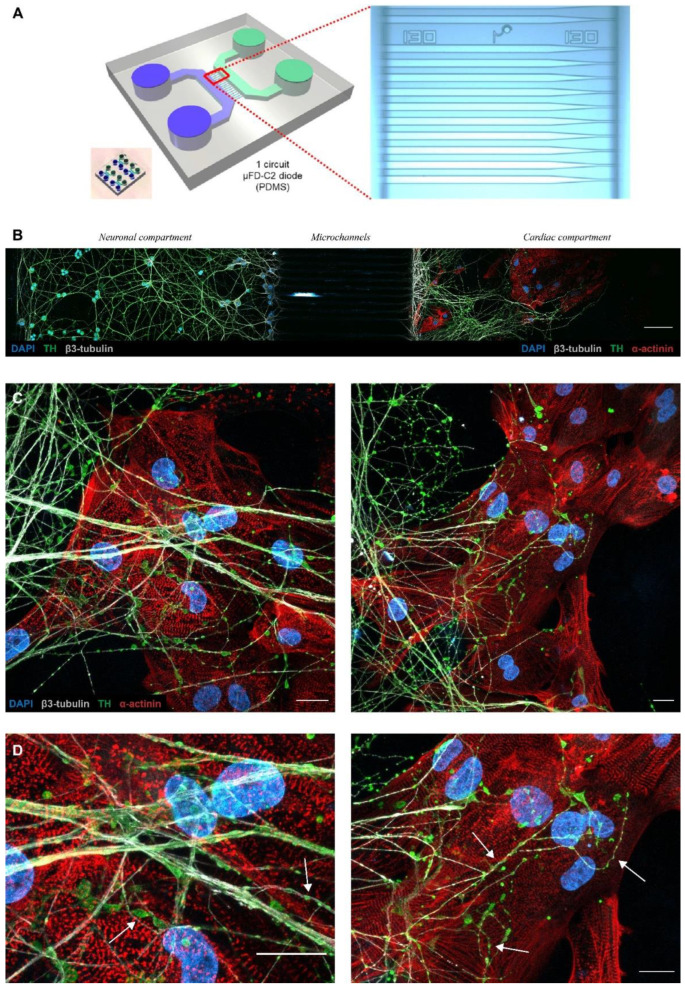
Neuronal diodes fluidic microsystems as tools to establish compartmentalized neurocardiac co-culture on chip. (**A**) Structural characteristics of a two-chamber neuronal diode also known as Brainies^®^. **Left panel**: 3D model of the micro-structured PDMS-based block (grey), including cylindric wells (diameter: 4 mm; height: 5 mm), rectangular cell culture chambers (vol. ~1 µL; height: 55 µm), and a series of asymmetrical microchannels (length: 500 µm; height: 3 µm; width: tapering from 15 µm to 3 µm). The compartment where neurons are seeded is represented in blue and the compartment where cardiomyocytes are seeded is represented in green. The small insert shows a picture of one of the Brainies^®^ that contains four neuronal diodes. **Right panel**: micro-graph of a portion of the funnel-shaped microchannels. (**B**) In this fluorescence microscopy image of a microfluidic chip section, the neural compartment is located to the left and the cardiac compartment to the right. Between the two compartments are the microchannels (not visible in the image, scale bar: 100 µm). PC12 cells differentiate directly into their compartments and their axons project into the cardiac compartment through the channels. (**C**,**D**) Images of neuro–cardiac interactions in cardiac compartments after 14 days of co-culture. Tyrosine hydroxylase (in Green) appear apposed along axons. Axons (β3-tubulin; in white) harbor varicosities in the interaction zones (white arrows). (Blue: DAPI, Red: sarcomeric α-actinin, green: Tyrosine hydroxylase, white: β3-tubulin, scale bar: 20 µm).

**Figure 3 cells-11-03764-f003:**
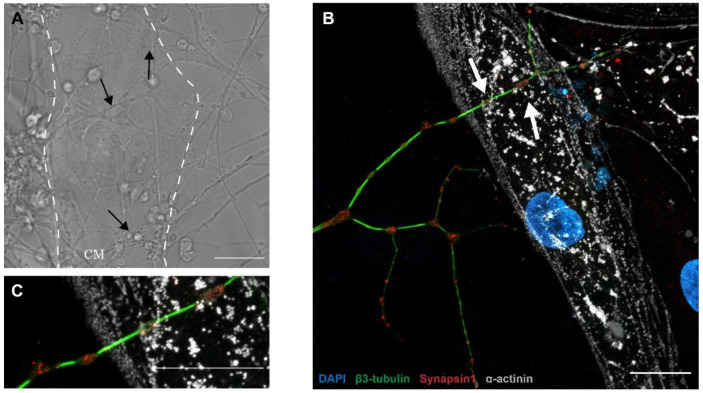
Structural characterization of the neuro–cardiac junction on a 14-day-old OOC. Images showing regions of local interaction between PC12 axons and hiPSC cm in transmission light ((**A**), black arrows) or in immunofluorescence (**B**). PC12 neurites express the protein synapsin-1 colocalized with structural staining (β3-tubulin; green, white arrows) in regions of interaction with cardiomyocytes and neurites. (**C**) Co-localization of tyrosine hydroxylase and synapsin-1 in the varicosities outside the neurocardiac junctions. (In this figure: blue: DAPI, red: synapsin-1, green: β3-tubulin, grey: sarcomeric α-actinin, scale bar: 20 µm.)

**Figure 4 cells-11-03764-f004:**
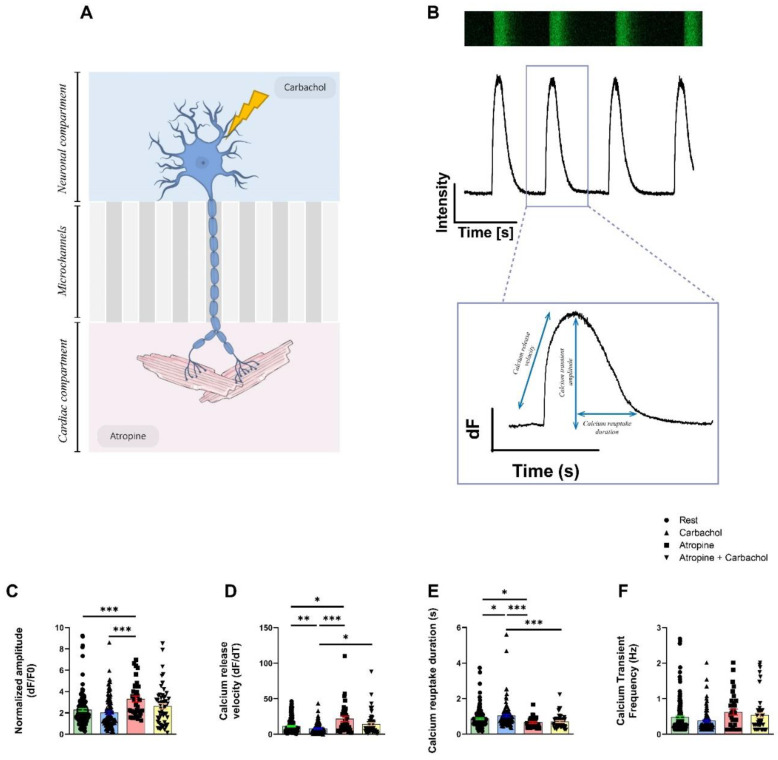
Monitoring of the intracellular calcium cycling properties on 14-day-old neuro–cardiac OOC using PC12 and hiPSC-CMs. (**A**) Schematic representation of neurocardiac OOCs with cellular compartments showing where the different molecules are added. The M2 muscarinic receptor antagonist Atropine is added to the cardiac compartment and the muscarinic receptor agonist carbachol is added to the neuronal compartment. (**B**) CaT are recorded using Fluo4-AM. Representation of CaT in hiPSC-CMs recorded in line-scan configuration with the corresponding plot below from which calcium kinetics are obtained. (**C**) Parameters of normalized amplitude of CaT, (**D**) calcium release velocity, (**E**) duration of Ca^2+^ reuptake and (**F**) frequency of CaT are presented in resting condition, after addition of 1.5 mM Carbachol and or 1 µM Atropine on the cardiomyocytes located in the cardiac compartment after 14 days of co-culture. Data are presented as mean ± SEM, Rest *n* = 145, Carbachol *n* = 108, Atropine *n* = 34, Carbachol + Atropine *n* = 55. Dunn’s multiple comparisons test were performed, * *p* < 0.05, ** *p* < 0.01, *** *p* < 0.001.

**Figure 5 cells-11-03764-f005:**
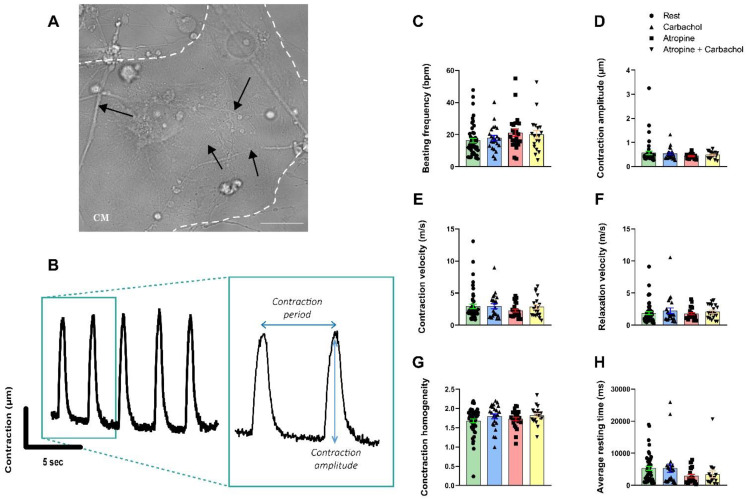
Contractile properties of hiPSC-CMs recorded in 14-day-old neuro–cardiac OOC by video-edge capture. (**A**) Picture of PC12 neurites connected (marked with black arrows) to hiPSC-CMs in transmitted light. Spontaneous contraction is recorded on cardiac compartment for 25 seconds. (**B**) Representation of contraction/relaxation cycle obtained from movies. (**C–H)** Measurement of contractile parameters on neuro–cardiac OOC at rest, in the presence of 1.5 mM of Carbachol, 1 µM of Atropine, or in the presence of both. (**C**) Beating frequency, (**D**) amplitude of contraction, (**E**) contraction and (**F**) relaxation velocity, (**G**) contraction homogeneity and (**H**) average resting time are obtained from series of images as shown in Figure 5A. Data are presented as mean ± SEM, Rest *n* = 49, Carbachol *n* = 23, Atropine *n* = 23, Carbachol + Atropine *n* = 19, Dunn’s multiple comparisons test were performed.

**Figure 6 cells-11-03764-f006:**
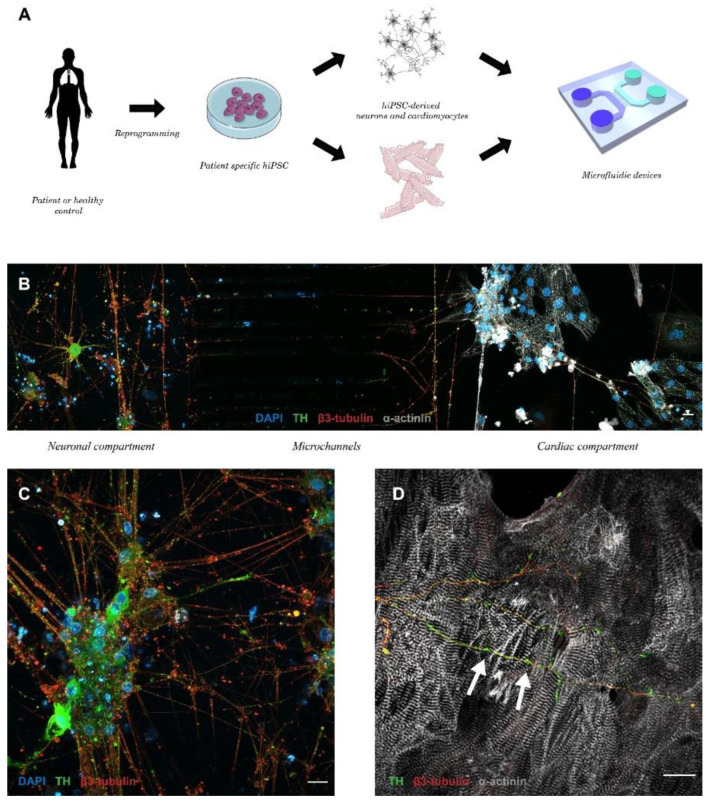
Neuro–cardiac OOC using neurons and cardiomyocytes derived from the same patient-derived hiPSC line. (**A**) Schematic presentation of the production of patient-specific neuro–cardiac OOC using similar control hiPSC line to generate autonomic neurons and cardiomyocytes. (**B**) Global view of neuro–cardiac interaction after 14 days in microfluidic chips of autonomic hiPSC-NRs seeded in neuronal compartment (**C**) projecting to the cardiac compartment. (**D**) Autonomic neurons expressing TH create projection that cross the microchannels and interact with cardiomyocyte (blue: DAPI, red: sarcomeric α-actinin, white: β3-tubulin, green: TH, scale bar: 20 µm).

## Data Availability

All material used here is distributed freely.

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
