# Peer review of "Impact of Neurons on Patient-Derived Cardiomyocytes Using Organ-On-A-Chip and iPSC Biotechnologies"

_cells, 2022, doi:10.3390/cells11233764_

Round 1

Reviewer 1 Report

The subject of this work is fascinating. The authors of the MS presented some helpful information and opened many questions. The technique used in a two-compartment organ-on-chip to culture neurons on one side and cardiomyocytes (CM) on another side with tapered channels. Such a device configuration is not new but it is interesting to see how neurons interact with CMs. The presented results are mostly based on rat PC12 cells and hiPSC-derived CMs, even though the penetration of hiPSC-neurons into the cardiac space has also been observed. The correlation between PC12-neurons and hiPSC CMs has been attributed to neuro-cardiac junctions with and without stimulation factors. Overall, this work is careful and the presented method is promising. Therefore, more discussions/information can be provided such as:  

1) The importance of this subject (both in the beginning and at the end)

2) Describe briefly the in vivo scenario of coupling, where the neural signal comes from, and how is stimulated 

3) Comparison of hiPSC-Cm beating with and without neurons

4) How the contraction/relaxation cycle varies with time after neural stimulation? 

5) Is hiPSC-CM permanently sensitive to the stimulation? 

6) Can CMs be modulated with both excitation and prohibition neural signals? 

Author Response

We thank the reviewers for their pertinent and fruitful comments that clearly improve our study. We below addressed each point and provided replies including new data illustrated by 3 new figures.

Reviewer 1:

“The subject of this work is fascinating. The authors of the MS presented some helpful information and opened many questions. The technique used in a two-compartment organ-on-chip to culture neurons on one side and cardiomyocytes (CM) on another side with tapered channels. Such a device configuration is not new but it is interesting to see how neurons interact with CMs. The presented results are mostly based on rat PC12 cells and hiPSC-derived CMs, even though the penetration of hiPSC-neurons into the cardiac space has also been observed. The correlation between PC12-neurons and hiPSC CMs has been attributed to neuro-cardiac junctions with and without stimulation factors. Overall, this work is careful and the presented method is promising. Therefore, more discussions/information can be provided such as :

1) The importance of this subject (both in the beginning and at the end)

Reply: We reinforced the importance of this topic in the introduction and discussion sections. Please see page 2.

2) Describe briefly the in vivo scenario of coupling, where the neural signal comes from, and how is stimulated

Reply: We described the in vivo scenario of coupling in the introduction section. Please see page 2.

3) Comparison of hiPSC-Cm beating with and without neurons

Reply: We generated a novel figure showing the kinetic properties of the intracellular calcium cycling in hiPSC-CMs cultivated alone or with PC12 in microfluidic devices for 14 days. We observed that PC12 cause an increase of the calcium release velocity. It suggests that PC12 increase the rising phase of the calcium transients. These data confirm that the neuronal extensions connected to the hiPSC-CMs have an impact on the SR calcium cycling. Please see the new supplemental figure 2. We described these results in the result section and discussed them in the discussion section. Please see pages 12 and 16.

4) How the contraction/relaxation cycle varies with time after neural stimulation? 

Reply: We do not have information about the variation of contraction cycle with time. But we know from different experiment that neurotransmitter may stay in medium for few minutes and stimulate their receptor.

For neurotransmitter quantification, the PC12 cells were stimulated with carbachol and medium was collected between 1 and 2 minutes after the stimulation. With this approach, we determined that neurotransmitter was not degraded within this duration. Moreover, functional properties of cardiomyocytes were recorded after neuronal stimulation for at least 10 minutes, showing that these effects can last this time.

5) Is hiPSC-CM permanently sensitive to the stimulation? 

Reply: Our neuro-cardiac OOC was made for up to 3 weeks following the co-cultures and connexion between the neurons and hiPSC-CMs. Unfortunately, we did not evaluate whether the hiPSC-CMs are still sensitive to the neuronal stimulation over the time of the co-culture. However, we could expect that, once the neuronal extensions reach the hiPSC-CMs, the latter are progressively sensitive to the neurons. As we observed that neurons increase their extensions and connexions with the hiPSC-CMs over time, we could expect a progressive chronological impact of the neurons on the hiPSC-CMs associated with further physical interactions.

6) Can CMs be modulated with both excitation and prohibition neural signals? 

Reply: We observed that PC12 cells express choline acetyltransferase, tyrosine hydroxylase, and dopamine β hydroxylase in the same neurite. Therefore, we hypothesized a combined release of acetylcholine and catecholamines at the synaptic level. To test it, we blocked the cholinergic system by adding atropine to the cardiac compartment. When carbachol and atropine were added together, we promoted the hiPSC-CM stimulation by sympathetic system. However, we did not observe any effect. These results may be explained by the low secretion of catecholamines by PC12 cells as we have quantified in Figure 1. We have reinforced these results in the discussion section. Please see pages 15 and 16.

Reviewer 2 Report

I would like to thank the authors for submitting this manuscript to Cells and for pursuing this work. Overall, this is an interesting approach: modeling the neuro-cardiac junction. I believe that this manuscript requires some additional work to support its conclusions: I recommend that the authors consider adding the appropriate controls to their experiments. Specifically, in figures 3, 4, and 6, they should develop monocultures (i.e., only cardiomyocytes or only neurons) and perform the same staining demonstrating that the structures they are describing are absent. Additionally, they should consider adding small arrows to denote the structures they are identifying. The authors should also perform the experiments described in figures 4 and 5 in monocultures as well to demonstrate that treatment in the absence of one of the two cellular components does not lead to the same outcome. Finally, I recommend that the authors consider expanding the section in the conclusions where they are comparing their technology/findings to what has been described previously in the literature. 

Author Response

We thank the reviewers for their pertinent and fruitful comments that clearly improve our study. We below addressed each point and provided replies including new data illustrated by 3 new figures.

Reviewer 2:

I would like to thank the authors for submitting this manuscript to Cells and for pursuing this work. Overall, this is an interesting approach: modeling the neuro-cardiac junction. I believe that this manuscript requires some additional work to support its conclusions: I recommend that the authors consider adding the appropriate controls to their experiments. Specifically, in figures 3, 4, and 6, they should develop monocultures (i.e., only cardiomyocytes or only neurons) and perform the same staining demonstrating that the structures they are describing are absent. Additionally, they should consider adding small arrows to denote the structures they are identifying. The authors should also perform the experiments described in figures 4 and 5 in monocultures as well to demonstrate that treatment in the absence of one of the two cellular components does not lead to the same outcome. Finally, I recommend that the authors consider expanding the section in the conclusions where they are comparing their technology/findings to what has been described previously in the literature.” 

Reply: As suggested by the reviewer 2, we generated control experiments including monocultures of hiPSC-CMs in OCC (i.e., in the absence of PC12 cells). Supplemental figure 3 shows the effect of carbachol applied in the neuronal compartment while atropine was applied in the cardiac compartment containing hiPSC-CMs. We found that atropine has no effect on the SR calcium handling kinetic properties. Carbachol did not change the calcium release velocity, calcium reuptake duration or calcium transient frequency (Supplemental figure 3). However, addition of carbachol surprisingly increases the calcium transient amplitude. These results suggest that in the absence of axons in the microchannels, carbachol may diffuse, over time, from the neuronal compartment to the cardiac compartment to modulate the SR calcium cycling of the hiPSC-CMs. Carbachol is a muscarinic receptor agonist. These results suggest that hiPSC-CMs are sensitive to carbachol. One study has demonstrated the ionotropic role of carbachol on guinea pig cardiomyocytes by increasing the calcium transients and contractions (PMID: 9435162). Overall, our results indicate that PC12 have an impact on the SR calcium handling not only limited to muscarinic receptor stimulation by acetylcholine. These functional results support our quantification of the neurotransmitters in the supernatant of PC12 cells (Figure 1). Very likely, the PC12 neurites release mixture of neurotransmitters including acetylcholine, dopamine and other catecholamines that lead to more complex consequences on the SR calcium handling properties. We described these results in the result section and discussed them in the discussion section. See pages 15 and 16 of the revised manuscript.

To reinforce our immunostaining results, we generated a new supplemental figure 1 with 30-days-old hiPSC-CMs alone in the OOC and stained with DAPI and synapsin-1a antibody. As expected, our results show that hiPSC-CMs do not express synapsin-1a and tyrosine hydroxylase in absence of PC12. Therefore, they prove that the synaptic junction, associated with synapsin-1a expression, is formed only when the neuronal extensions reach the hiPSC-CMs in the OOC (new supplemental figure 1). We described these results in the result section. Please see page 9 of the revised manuscript.

We have added small arrows to denote the structures in Figures 2, 3 and 6 and have updated the legend accordingly.

Round 2

Reviewer 2 Report

The authors have adequately addressed my previous feedback.